# Construction of a Heterotrophic Nitrification–Aerobic Denitrification Composite Microbial Consortium and Its Bioaugmentation Role in Wastewater Treatment

**DOI:** 10.3390/biology14121734

**Published:** 2025-12-04

**Authors:** Wenjing Jiao, Haoyang Sun, Zixuan Zhang, Zuyin Xiao, Hanhan Song, Jiale Liu, Xiaole Xu, Juan Wang, Guiying Wang, Jiang Zhang, Chenyang Wang, Lusheng Li, Lifei Chen

**Affiliations:** 1Shandong Province Engineering Research Center of Black Soldier Fly Breeding and Organic Waste Conversion, College of Agriculture and Biology, Liaocheng University, Liaocheng 252000, China; 2College of Life Science and Technology, Xinjiang University, Urumqi 830049, China

**Keywords:** heterotrophic nitrification-aerobic denitrification, composite microbial consortium, sequencing batch reactor (SBR), bioaugmentation, microbial community, wastewater treatment

## Abstract

To address wastewater nitrogen pollution and the limitations of traditional treatment methods, this study constructed a heterotrophic nitrification–aerobic denitrification (HN-AD) composite microbial consortium using three pre-screened strains: *Delftia tsuruhatensis* SDU2, *Pseudomonas stutzeri* SDU10, and *Alcaligenes faecalis* SDU20. An orthogonal test optimized their inoculation ratio to 2:3:3, achieving an in vitro ammonium removal efficiency of 96.02%. In sequencing batch reactor (SBR) experiments, the bioaugmented reactor (SBR1) outperformed the control (SBR2) in pollutant removal. By day 40, SBR1 reached 88.9% ammonium removal and 93.7% COD removal, 20.5% and 17.9% higher than SBR2, respectively. Microbial community analysis showed bioaugmentation enriched phyla like Proteobacteria and Bacteroidota, as well as functional genera such as *Alcaligenes*, which synergized with indigenous microbiota to enhance efficiency. This HN-AD consortium offers a promising strategy for optimizing wastewater treatment.

## 1. Introduction

Nitrogen pollution in wastewater remains a pressing environmental concern, prompting the need for efficient and sustainable treatment technologies [1,2,3,4]. With the development of industrialization and agricultural intensification, excessive nitrogen enters water bodies through various channels, leading to eutrophication of water bodies, imbalance of the ecosystem, and a drinking water safety crisis [5,6]. It has become a major environmental issue restricting sustainable development. Traditional methods for treating nitrogen pollution in water bodies primarily include physical approaches (e.g., reverse osmosis, adsorption), chemical methods (e.g., breakpoint chlorination, chemical precipitation), and biological techniques (e.g., conventional nitrification–denitrification, constructed wetlands) [4,7,8,9,10,11,12]. However, these methods often suffer from inefficiency, high costs, or secondary pollution. For instance, physical methods are energy-intensive and prone to membrane fouling, while chemical approaches generate toxic byproducts and involve costly reagents [13,14,15,16,17]. Conventional biological nitrogen removal relies on two distinct processes: nitrification and denitrification. Nitrification refers to the oxidation of ammonium nitrogen (NH_4_^+^-N) into nitrite (NO_2_^−^-N) and nitrate (NO_3_^−^-N) by autotrophic nitrifying bacteria under aerobic conditions. Denitrification involves the reduction of nitrate or nitrite into nitrogen gas (N_2_) by heterotrophic denitrifying bacteria under anoxic/anaerobic conditions. However, since nitrifying and denitrifying bacteria are functionally distinct microbial groups with conflicting environmental requirements—autotrophic nitrifying bacteria (obligate aerobes) are slow-growing, oxygen-dependent autotrophs sensitive to environmental fluctuations, whereas heterotrophic denitrifying bacteria (facultative anaerobes) require organic carbon as electron donors and are oxygen-sensitive—they cannot coexist or synergize in the same reactor simultaneously without alternating operating conditions (e.g., BIODENIPHO technology) [14,15,18,19,20,21,22,23]. This spatial and temporal separation increases operational complexity and costs (e.g., additional anoxic tank or extra mixing energy) [9,18,19,21,23,24,25,26].

In recent years, novel biological nitrogen removal technologies have continuously emerged, such as partial nitrification-denitrification, anaerobic ammonium oxidation (Anammox), and simultaneous nitrification-denitrification (SND), offering innovative strategies to overcome the limitations of traditional methods [1,11,27,28,29,30,31]. These advancements enable more efficient nitrogen removal through reduced energy consumption, minimized carbon footprint, and simplified reactor configurations, paving the way for sustainable wastewater treatment in scenarios with low carbon-to-nitrogen ratios, complex industrial effluents, or stringent environmental regulations. The discovery of HN-AD microorganisms has enabled SND, emerging as a groundbreaking focus in the field of biological nitrogen removal [10,32,33,34,35,36]. These HN-AD microbes can perform both nitrification (converting ammonium to nitrate/nitrite) and denitrification (reducing nitrate/nitrite to nitrogen gas) within a single reactor under aerobic conditions. This integration eliminates the need for separate reaction zones, significantly shortens treatment cycles, and reduces spatial footprint and infrastructure costs [37,38]. Moreover, HN-AD microorganisms exhibit robust environmental resilience (e.g., tolerance to fluctuating dissolved oxygen, pH, and salinity), effectively addressing the limitations of conventional biological nitrogen removal systems, such as slow growth rates of autotrophic nitrifiers and strict carbon/oxygen requirements for denitrifiers [15,23,39,40,41,42,43,44,45]. Their ability to thrive in diverse conditions makes them a promising solution for treating complex wastewater streams, particularly those with low carbon-to-nitrogen ratios or high operational variability [46,47].

In recent years, significant progress has been made in the study of HN-AD microorganisms. Researchers worldwide have isolated diverse high-efficiency HN-AD strains from environments such as activated sludge in wastewater treatment plants, marine sediments, and river sludge, including bacterial genera like *Pseudomona* [4,5,48,49,50,51,52,53], *Bacillus* [32,54,55,56], and *Acinetobacter* [7,47,57], as well as fungal species such as the red yeast *Sporidiobolus pararoseus* Y1 [58]. Studies have revealed two distinct nitrogen removal pathways in HN-AD microorganisms: one follows the conventional nitrification-denitrification sequence (NH_3_ → NO_2_^−^ → NO_3_^−^ → N_2_), while the other directly converts ammonia to nitrogen gas via hydroxylamine (NH_2_OH) intermediates, bypassing nitrite and nitrate accumulation [49,50,59]. However, the nitrogen removal performance of these strains is highly influenced by factors such as carbon source type, carbon-to-nitrogen (C/N) ratio, temperature, pH, and dissolved oxygen (DO), with significant metabolic variations observed even among strains of the same genus. Although laboratory-scale studies have demonstrated the efficacy of single HN-AD strains (e.g., *Bacillus subtilis* A1 and *Vibrio diabolicus* SF16) in enhancing nitrogen removal in sequencing batch reactors (SBRs) and biological aerated filters (BAFs), achieving total nitrogen (TN) removal rates of up to 81.3% and 73.92%, research on complex wastewater (e.g., high ammonia, low C/N ratio) remains limited [60,61]. Furthermore, the synergistic effects of multi-strain HN-AD consortia and their impact on the microbial community structure of activated sludge are poorly understood. To address these gaps, this study focuses on constructing tailored microbial consortia using pre-screened HN-AD strains validated for swine wastewater treatment. By employing SBR systems, we investigate the bioaugmented nitrogen removal performance of these consortia and their influence on the indigenous microbial community dynamics, aiming to advance the transition of HN-AD technology from laboratory research to engineered applications.

## 2. Materials and Methods

### 2.1. Strain

The strains *Delftia tsuruhatensis* SDU2 [62], *Pseudomonas stutzeri* SDU10 [63], and *Alcaligenes faecalis* SDU20 [64] were previously isolated from activated sludge of a swine wastewater treatment plant and stored in 30% glycerol tubes at –80 °C.

### 2.2. Medium

Luria–Bertani medium (LB): 25.0 g/L LB medium mixed powder, 1 L distilled water; enrichment medium: 0.2 g/L NaNO_2_, 5.0 g/L CaCO_3_, 0.5 g/L NaCl, 0.5 g/L K_2_HPO_4_, 0.5 g/L MgSO_4_·7H_2_O, 0.4 g/L FeSO_4_·7H_2_O, 1 L distilled water; screening isolation medium (VM): 2.0 g/L acetamide, 8.2 g/L KH_2_PO_4_, 1.6 g/L NaOH, 0.5 g/L MgSO_4_·7H_2_O, 0.5 g/L KCl, 0.0005 g/L CaSO_4_·2H_2_O, 0.0005 g/L CuSO_4_·5H_2_O, 0.0005 g/L FeCl_3_·6H_2_O, 0.0005 g/L ZnSO_4_·H_2_O, 1 L distilled water (weighed using an analytical balance with 0.1 μg precision); nitrification medium (BM): 5.6 g/L disodium succinate, 1.5 g/L KH_2_PO_4_, 0.47 g/L (NH_4_)_2_SO_4_, 7.9 g/L Na_2_HPO_4_·7H_2_O, 0.1 g/L MgSO_4_·7H_2_O, 2 mL/L trace element solution, 1 L distilled water; trace element solution: 50.0 g/L Na_2_EDTA, 2.2 g/L ZnSO_4_·7H_2_O, 5.5 g/L CaCl_2_, 5.06 g/L MnCl_2_·4H_2_O, 5.0 g/L FeSO_4_, 1.57 g/L CuSO_4_·5H_2_O, 1.60 g/L CoCl_2_·6H_2_O, 1 L distilled water; All reagents were purchased from Macklin regent (Shanghai, China).

### 2.3. Experimental Setup and Design

This experiment utilized two identical SBR systems constructed from acrylic polymer (PMMA) with an inner diameter of 80 mm, height of 1000 mm, total volume of 5 L, and effective volume of 4 L, each externally equipped with a 120 mm-diameter insulation jacket (height 1000 mm) to maintain thermostatic control at 30 °C via immersion heating elements in a water bath. The fully automated 8 h operational cycle, governed by timer-controlled solenoid valves, sequentially executed five phases (Figure 1): Fill phase (10 min): 3 L influent loading via peristaltic pump at 0.3 L/min through silicone tubing from the reactor base; Aeration phase (430 min): Microporous diffuser-mediated oxygenation using an air compressor with mass flow controller-regulated airflow (Changzhou Shuanghuan Thermal Engineering Instruments Co., Ltd., Changzhou, China) to ensure homogeneous dissolved oxygen distribution and sludge floc fluidization; Sedimentation phase (30 min): Static sedimentation; Decant phase (5 min): Effluent discharge through an outlet 200 mm above reactor base via normally-closed solenoid valves; Idle phase (5 min). Sampling ports were positioned 400 mm from the reactor bottom for process monitoring. Evaporation was accounted for by regular top-ups with sterile distilled water to maintain the effective volume, and no condenser was used.

### 2.4. Construction of Synthetic Microbial Consortia

Strains SDU2, SDU10, and SDU20 were individually cultured in LB medium to an optical density of OD600 = 1, followed by cell density standardization to a uniform concentration of 1 × 10^9^ CFU/mL via centrifugation (4000× *g*, 10 min) and resuspension in 0.85% (*w*/*v*) sterile saline (0.145 M NaCl, autoclaved at 121 °C for 20 min). The standardized suspensions were then inoculated into sterilized nitrification medium at defined volumetric ratios. The inoculation ratios were optimized through a three-factor, three-level orthogonal experimental design (L9(34) matrix), where factors A, B, and C represented strains SDU2, SDU10, and SDU20, respectively, with inoculation volumes set at 1 mL, 2 mL, and 3 mL (see Table 1). Each consortium was incubated under aerobic conditions (30 °C, 150 rpm) for 24 h, after which final ammonium concentration was quantified via Nessler’s reagent spectrophotometry (λ = 420 nm), and ammonium removal efficiency (η, %) was calculated as: η = C0/(C0 − Ct) × 100%. where C0 and Ct denote initial and final ammonium concentrations (mg/L).

### 2.5. Bioaugmentation Efficacy and Microbial Diversity Assessment of Synthetic Microbial Consortia in Sequencing Batch Reactors

#### 2.5.1. SBR Operation Phases

The bioaugmented nitrogen removal experiment was conducted using the microbial consortium constructed in Section 2.4. Two identical SBRs—designated as the experimental group (SBR1) and control group (SBR2)—were inoculated with activated sludge (sludge concentration: 4390 mg TSS/L) collected from the aerobic tank of an A^2^/O process at the domestic wastewater treatment facility of Shandong University (Qingdao Campus). During the start-up phase, 30% of the reactor volume was filled with the inoculated sludge. Both reactors were fed exclusively with synthetic wastewater (composition detailed in Table 2, pH: 7.2 ± 0.2, ORP: +150~+200 mV, dissolved oxygen: 2~4 mg/L) and operated through four sequential phases: start-up, acclimation, stabilization, and bioaugmentation. In the first three phases, SBR1 and SBR2 operated synchronously. At the onset of the bioaugmentation phase, the pre-constructed microbial consortium was introduced into SBR1 at a dosage equivalent to 50% of the influent volume, while SBR2 remained unaltered without exogenous microbial addition. Daily influent and effluent water quality parameters were monitored throughout the experiment. Post-reactor initiation, each SBR operated with three cycles per day (8 h/cycle), comprising 10 min of feeding, 430 min of aerobic reaction, 30 min of sedimentation, 5 min of decanting, and 5 min of idle time, ensuring precise control of hydraulic retention time (HRT) and sludge retention time (SRT).

#### 2.5.2. 16S rRNA Amplicon Sequencing and Data Processing

To evaluate the influence of exogenous synthetic microbial consortia on the microbial community structure of activated sludge, samples were collected from SBR1 and SBR2 during four operational phases: start-up phase (A1, A2), acclimation phase (days 1–15) (B1, B2), stabilization phase (days 16–30) (C1, C2), and enhancement phase (days 31–40) (D1, D2). Total genomic DNA was extracted using the FastDNA Spin Kit for Soil (MP Biomedicals, Santa Ana, CA, USA), followed by PCR amplification of the V3-V4 hypervariable regions of the 16S rRNA gene using primers 338F (5′-ACTCCTACGGGAGGCAGCAG-3′) and 806R (5′-GGACTACHVGGGTWTCTAAT-3′). PCR thermocycling profile: Initial denaturation: 95 °C for 3 min; 35 cycles: 95 °C for 30 s, 55 °C for 30 s, 72 °C for 45 s; Final extension: 72 °C for 10 min. Amplicons were sequenced on the Illumina MiSeq (Illumina Inc., San Diego, CA, USA) platform (2 × 300 bp paired-end). by Shanghai Majorbio Bio-pharm Technology Co., Ltd. (Shanghai, China). Bioinformatics processing (including OTU clustering, alpha/beta diversity analysis, and taxonomic classification) was performed on the Majorbio Cloud Platform (http://www.majorbio.com). Raw sequencing data were denoised using exact sequence variants [65]. Chimeric sequences were removed using USEARCH 7.0 based on the SILVA database [66]. Operational taxonomic units (OTUs) were determined using BLAST 2.12.0 by searching for representative sequences against the SILVA database v132 using the ‘best hit’ approach, 99% similarity (ZOTUs) for species-level resolution [67]. The community richness index (Chao1) and community diversity indices (Simpson and Shannon) used to estimate α-diversity were calculated using QIIME 2 and visualised using R software v3.2.0 (https://cran.r-project.org/src/base/R-3/ accessed on 27 December 2024). The relative abundance of the order is displayed in a stacked column. For β-diversity analysis, principal coordinate analysis (PcoA) based on weighted UniFrac distance was used to visualise the OTU data. Taxonomic cladograms were constructed based on the OTUs. SPSS software (version 22.0; SPSS Inc., Chicago, IL, USA) was used for comprehensive and systematic statistical analyses of all monitored test data. Raw sequencing data were deposited in the NCBI Sequence Read Archive (SRA) under BioProject accession number PRJNA778940.

### 2.6. Data Analysis

The concentrations of ammonium nitrogen (NH_4_^+^-N), nitrate nitrogen (NO_3_^−^-N), and nitrite nitrogen (NO_2_^−^-N) were determined daily using standardized methods as outlined, specifically Nessler’s reagent spectrophotometry (λ = 420 nm; APHA 4500-NH_3_ D) for NH_4_^+^-N, ultraviolet spectrophotometric screening (λ = 220/275 nm; APHA 4500-NO_3_^−^ B) for NO_3_^−^-N, and N-(1-naphthyl)ethylenediamine dihydrochloride (NED) colorimetry (λ = 540 nm; APHA 4500-NO_2_^−^ B) for NO_2_^−^-N [62,63,64]. Chemical oxygen demand (COD)was measured with a Hach DR3900 multiparameter water quality analyzer (Hach Company, Loveland, CO, USA) employing TNTplus™ COD test vials (Method 8000, range 3–150 mg/L), in compliance with USEPA Method 410.4 and ISO 6060:1989 [62,68]. All analyses were performed in triplicate, and data are presented as mean ± standard deviation (SD). Statistical analyses included Linear Mixed Model (LMM) for repeated-measures data, LefSe analysis (LDA > 2.0, FDR-adjusted *p* < 0.05) for taxonomic differences, Kruskal–Wallis test with Benjamini–Hochberg correction for genus-level comparisons, and principal coordinate analysis (PcoA) based on weighted UniFrac distance for β-diversity. The heatmap was generated using R software v3.2.0, with data normalized via variance-stabilizing transformation to account for differences in sequencing depth.

## 3. Results

### 3.1. Construction of Synthetic Microbial Consortia

Strains SDU2, SDU10, and SDU20 were combined according to the volumetric ratios defined in the orthogonal experimental matrix (Table 3). Each experimental group was incubated for 24 h, followed by measurement of ammonium removal efficiency (η, %). Results are summarized in Table 3.

R-value (Range) reflects the magnitude of each factor’s influence, where higher values indicate greater significance. Strain SDU20 (*A. faecalis*) exhibited the strongest impact on ammonium removal (R = 12.49), followed by SDU10 (*P. stutzeri*, R = 5.66) and SDU2 (*D. tsuruhatensis*, R = 2.70). The combination A2B3C3 yielded the highest ammonium removal efficiency. Specifically: *D. tsuruhatensis* SDU2: 2 mL; *P. stutzeri* SDU10: 3 mL; *A. faecalis* SDU20: 3 mL. Thus, the optimal volumetric ratio for the synthetic consortium is 2:3:3 (SDU2:SDU10:SDU20).

### 3.2. Impact of Synthetic Microbial Consortia on Pollutant Removal in SBRs

During the acclimation phase (Days 1–15), both reactors exhibited parallel trends in effluent COD reduction: SBR1 decreased from 1504 ± 52 mg/L (Day 1) to 543 ± 31 mg/L (Day 15), and SBR2 decreased from 1480 ± 47 mg/L (Day 1) to 552 ± 29 mg/L (Day 15). In the stabilization phase (Days 16–30), effluent COD stabilized at 484.3 ± 25 mg/L (76.6% removal efficiency) for SBR1 and 473.3 ± 28 mg/L (74.8% removal efficiency) for SBR2. Following bioaugmentation (Day 30), SBR1 achieved superior COD removal, with effluent concentrations dropping to 124 ± 18 mg/L (93.7% efficiency) by Day 40, compared to 494 ± 32 mg/L (75.8% efficiency) in SBR2 (Figure 2). This represents a 17.9 percentage-point improvement in COD removal, attributable to the consortium’s metabolic utilization of organic carbon for growth and degradation.

The variations in effluent ammonium (NH_4_^+^-N) concentrations across operational phases for SBR1 and SBR2 are shown in Figure 3. During the acclimation phase (Days 1–15), both reactors exhibited a continuous decline in NH_4_^+^-N: SBR1 decreased from 75.3 mg/L (Day 1, 27.9% removal efficiency) to 35.2 mg/L (Day 15, 67.8% efficiency), while SBR2 decreased from 74.2 mg/L (Day 1, 28.9% efficiency) to 33.9 mg/L (Day 15, 69.0% efficiency), demonstrating near-identical treatment performance and parallel trends with COD removal. In the stabilization phase (Days 16–30), average effluent NH_4_^+^-N concentrations stabilized at 32.6 mg/L (67.4% efficiency) for SBR1 and 33.3 mg/L (66.8% efficiency) for SBR2, confirming consistent nitrogen removal capacity. Following bioaugmentation of SBR1 with the synthetic consortium at the end of stabilization (Day 30), effluent NH_4_^+^-N in SBR1 progressively declined, reaching 11.4 mg/L (88.9% efficiency) by Day 40, whereas SBR2 (control) maintained similar levels to the stabilization phase (32.5 mg/L, 68.4% efficiency). This represents a 20.5% increase for SBR1 in ammonium removal efficiency compared to SBR2 (68.4% → 88.9%).

### 3.3. Impact of Nitrogen-Removal Bioaugmentation on Microbial Community Structure in SBR Reactors

#### 3.3.1. Analysis of Bacterial Community Diversity in SBR Sludge Across Operational Phases

Sequencing and optimization of the 16S rRNA gene V3-V4 hypervariable regions (Appendix A) in sludge samples from different operational phases of SBR reactors yielded a total of 541,265 sequencesacross eight samples (range: 60,774–74,078 sequences per sample; Table 4). Sequences were clustered into 1417 operational taxonomic units (OTUs)at 99% similarity, with taxonomic annotation revealing 37 phyla, 102 classes, 230 orders, 364 families, 602 genera, and 955 species.

As illustrated in Table 5, the alpha diversity indices (Shannon, Simpson, ACE, Chao1, coverage, and Smith-Wilson) of microbial communities in both SBRs showed minimal differences during the start-up, acclimation, and stabilization phases, attributable to identical operational conditions (e.g., synchronized hydraulic retention time, organic loading rate, and dissolved oxygen levels) that maintained comparable microbial diversity. However, during the enhancement phase, significant divergence emerged: SBR1 (bioaugmented with exogenous microbial consortia, sample D1) exhibited markedly higher Shannon (3.33 vs. 3.06), ACE (785.43 vs. 577.55), and Chao1 (808.68 vs. 563.89) indices compared to the control SBR2 (sample D2), alongside a reduced Simpson index (0.128 vs. 0.137). These metrics collectively indicate that bioaugmentation enriched species richness and evenness in SBR1, restructuring the microbial community through niche occupation or synergistic interactions with indigenous taxa. The observed 20% increase in ammonium removal efficiency further corroborates the functional impact of enhanced diversity. The findings underscore the critical role of exogenous consortia in modulating microbial ecology for optimized wastewater treatment.

#### 3.3.2. Temporal Shifts in Bacterial Community Composition at the Phylum Level in SBR Activated Sludge

As depicted in Figure 4, the dominant phyla (Proteobacteria, Bacteroidota, Actinobacteriota) align with typical activated sludge microbiomes in wastewater treatment plants, confirming the ecological relevance of the observed community structure. During the start-up phase, samples A1 and A2 exhibited nearly identical phylum-level compositions and closely matched relative abundances. Similarly, in the acclimation phase, samples B1 and B2 showed minimal structural divergence, reflecting uniform selective pressures under identical operational conditions. In the stabilization phase, samples C1 and C2 retained comparable phylum profiles but displayed slight variations in relative abundances (e.g., Proteobacteria at 58.2% in C1 vs. 54.9% in C2). The enhancement phase revealed marked divergence: in bioaugmented SBR1 (sample D1), Proteobacteria dominated at 66.5%, followed by Bacteroidota (21.7%), whereas control SBR2 (sample D2) was enriched with Bacteroidota (43.5%) and Proteobacteria (34.6%). This restructuring highlights the impact of exogenous consortia in driving niche competition and metabolic specialization, consistent with bioaugmented nitrogen removal systems where Proteobacteria dominance correlates with enhanced nitrification-denitrification activity.

As shown in Figure 5, the genus-level bacterial communities in SBR1 and SBR2 displayed minimal divergence within each operational phase (start-up, acclimation, and stabilization), yet exhibited pronounced structural shifts across these stages. During acclimation (samples B1 and B2), the relative abundances of denitrifying genera—including *Exiguobacterium*, *Stappia*, *Paracoccus*, and *Pseudomonas*—increased substantially relative to start-up (samples A1 and A2), mirroring the rapid decline in effluent ammonium concentrations. Genera such as *Alcaligenes* and *Pseudomonas* are known to mediate heterotrophic nitrification-aerobic denitrification [4,19,48,49,63,64]. Stabilization (samples C1 and C2) was characterized by the dominance of *Arcobacter*, *Peptococcaceae*, *Comamonas*, and *Aeromonas*, reflecting phase-specific metabolic adaptations. Metabolic adaptations included upregulation of functional genes related to ammonium monooxygenase (amoA) and nitrite reductase (nirS), inferred from taxonomic composition [4,19]. However, we acknowledge that amplicon data alone cannot fully confirm metabolic activity, and future studies will integrate metatranscriptomic analyses to validate these findings. During the enhancement phase, bioaugmentation of SBR1 with exogenous consortia triggered marked community restructuring, wherein *Alcaligenes*, *Citrobacter*, *Pseudomonas*, *Acidovorax*, and *Paracoccus* emerged as dominant taxa at significantly higher abundances than in the control SBR2. This shift indicates that the introduced consortia successfully enriched aerobic denitrifier populations while synergizing with indigenous microbiota, corresponding to substantial performance gains in SBR1, with ammonium and COD removal efficiencies reaching 88.9% and 93.7%, respectively. These findings support the hypothesis that exogenous consortia drive ecological niche partitioning, enhancing nitrogen removal through functional guild enrichment and metabolic pathway enhancement.

## 4. Discussion

The current study highlights the promising application of a synthetic HN-AD microbial consortium in SBR systems for enhanced nitrogen and organic pollutant removal. The optimized consortium—composed of *Delftia tsuruhatensis* SDU2, *Pseudomonas stutzeri* SDU10, and *Alcaligenes faecalis* SDU20 in a 2:3:3 ratio—enabled SBR1 to demonstrate superior performance. The reactor achieved removal efficiencies of 88.9% for ammonia nitrogen and 93.7% for COD, surpassing the performance of SBR2 by 20.5% and 17.9%, respectively. These findings align with previous research demonstrating the superiority of multi-strain HN-AD consortia over single-strain inoculations in terms of metabolic versatility and environmental resilience [69].

The enhanced performance of the bioaugmented reactor (SBR1) can be attributed to the synergistic interactions among the introduced strains and the indigenous microbial community. Microbial community analysis revealed that bioaugmentation enriched functional genera such as *Alcaligenes*, *Pseudomonas*, *Paracoccus*, and *Citrobacter*, all of which are known for their nitrogen transformation capabilities [4,19,39,54]. The dominance of Proteobacteria in SBR1, particularly during the enhancement phase, is consistent with previous studies that associate this phylum with efficient nitrification and denitrification activities. This shift in community structure suggests that the introduced consortium not only survived but also integrated into the native microbial ecosystem, enhancing its metabolic capacity [69].

The HN-AD pathway offers a significant advantage over conventional biological nitrogen removal processes by enabling simultaneous nitrification and denitrification under aerobic conditions [38]. This eliminates the need for separate anoxic and aerobic zones, simplifying reactor design and reducing operational costs. Moreover, the HN-AD microbes used in this study exhibited tolerance to fluctuating environmental conditions, making them suitable for treating complex wastewaters with low C/N ratios or variable influent characteristics [5]. These traits are particularly valuable in real-world applications where wastewater composition can be highly dynamic.

Despite the promising results, several challenges remain for the full-scale application of HN-AD consortia. First, the long-term stability of the introduced strains in open systems needs further investigation. While the consortium showed strong performance over a 40-day enhancement phase, it is unclear whether these benefits would persist under continuous operation or in the presence of inhibitory compounds commonly found in industrial wastewaters [69]. Second, the scalability of this approach needs to be validated in pilot-scale or full-scale reactors, where hydraulic and mass transfer limitations may affect microbial activity [33]. Additionally, the cost-effectiveness of large-scale inoculation and the potential ecological risks of introducing non-native strains should be carefully evaluated [70].

Future research should focus on optimizing the delivery and retention strategies of the consortium, such as immobilization techniques or granular sludge systems, to enhance microbial stability and reusability [70]. Moreover, integrating omics-based approaches (e.g., metagenomics, metatranscriptomics) could provide deeper insights into the functional dynamics of the consortium and its interactions with indigenous microbes [71]. Exploring the application of HN-AD consortia in treating specific industrial effluents, such as those from livestock farming or landfill leachate, could also broaden their practical utility [72].

## 5. Conclusions

This research successfully constructed an efficient HN-AD microbial consortium through orthogonal optimization, demonstrating its robust performance in enhancing nitrogen and organic matter removal in SBR systems. The optimal consortium (SDU2:SDU10:SDU20 at 2:3:3) significantly improved ammonium and COD removal efficiencies under aerobic conditions, attributed to synergistic interactions between introduced strains and indigenous microbes. Microbial community analysis revealed that bioaugmentation restructured the sludge microbiome, enriching functional taxa associated with nitrification and denitrification, which correlated with improved treatment performance. These findings provide empirical evidence for the practical application of HN-AD consortia in wastewater treatment, offering a sustainable solution to address nitrogen pollution challenges. Future studies could explore scaling up this technology for real-world complex wastewaters and investigating long-term operational stability.

## Figures and Tables

**Figure 1 biology-14-01734-f001:**
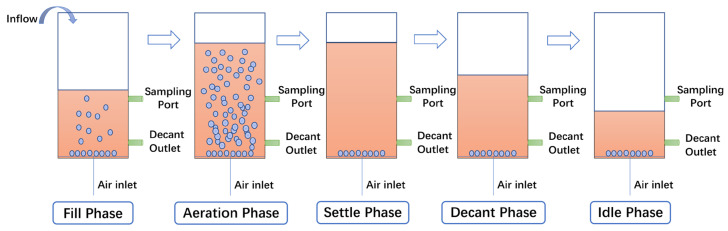
Operation flowchart of SBR reactor.

**Figure 2 biology-14-01734-f002:**
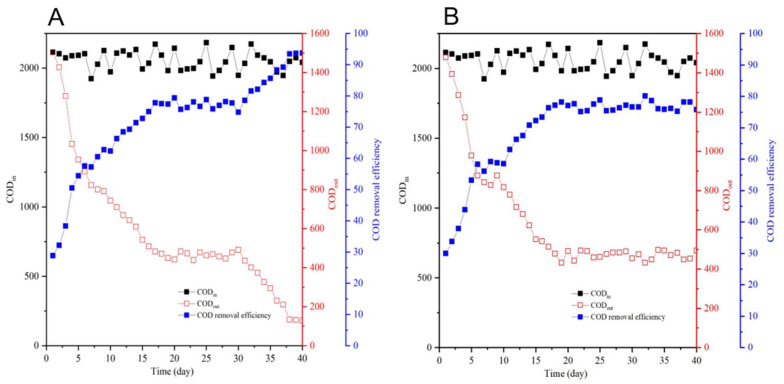
COD concentrations (left *y*-axis) and removal efficiencies (right *y*-axis) in SBR1 (**A**) and SBR2 (**B**) across operational phases. Error bars represent mean ± SD (*n* = 3).

**Figure 3 biology-14-01734-f003:**
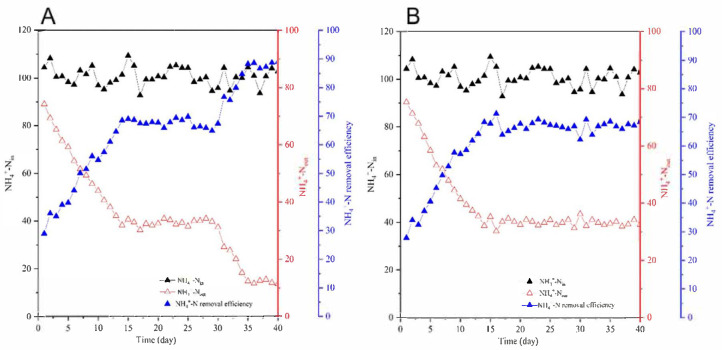
Effluent NH_4_^+^-N concentrations and removal efficiencies in SBR1 (bioaugmented) and SBR2 (control) across operational phases. (**A**), SBR1; (**B**), SBR2.

**Figure 4 biology-14-01734-f004:**
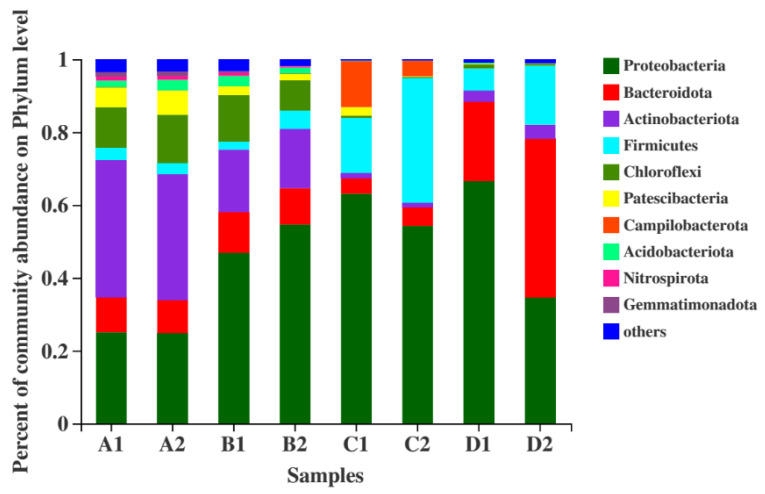
Relative abundance of dominant bacterial phyla in SBR sludge across operational phases. Error bars represent mean ± SD (*n* = 3).

**Figure 5 biology-14-01734-f005:**
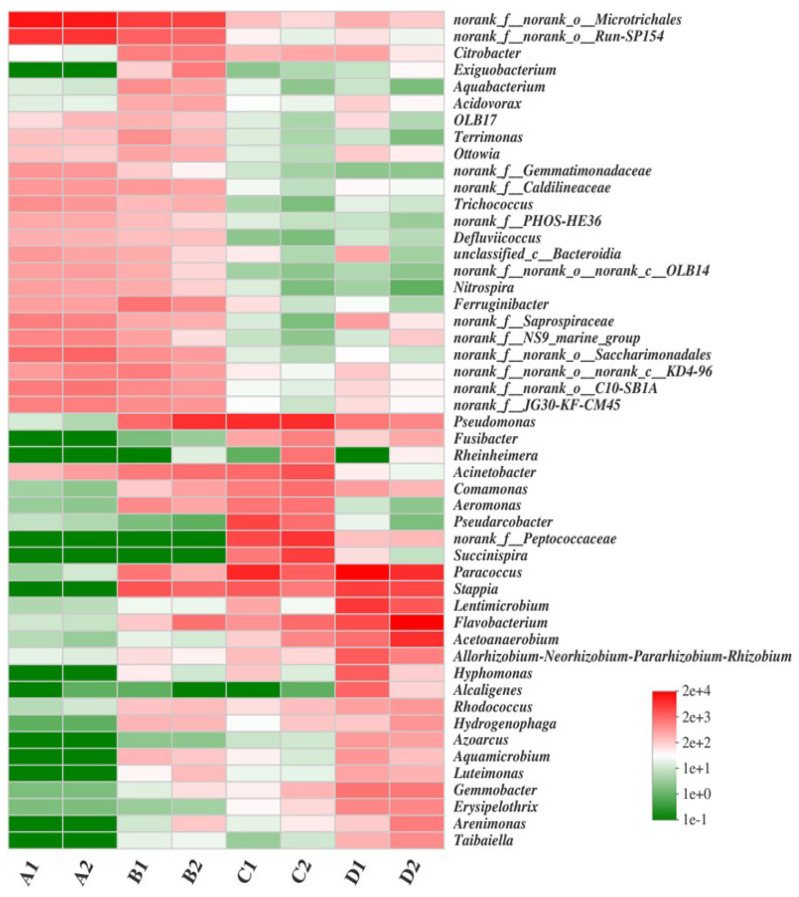
Heatmap of dominant bacterial genera (top 30) in SBR sludge. Color intensity indicates relative abundance (log-transformed). Data were normalized via variance-stabilizing transformation.

**Table 1 biology-14-01734-t001:** Factors and levels of orthogonal test.

Level	A	B	C
1	1 mL	1 mL	1 mL
2	2 mL	2 mL	2 mL
3	3 mL	3 mL	3 mL

Note: A = *Delftia tsuruhatensis* SDU2, B = *Pseudomonas stutzeri* SDU10, C = *Alcaligenes faecalis* SDU20.

**Table 2 biology-14-01734-t002:** Composition of Synthetic Wastewater.

Component	Concentration (g/L)
CH_3_COONa (Sodium acetate)	3.14
(NH_4_)_2_SO_4_ (Ammonium sulfate)	0.48
NaHCO_3_ (Sodium bicarbonate)	7.9
KHCO_3_ (Potassium bicarbonate)	1.5
MgSO_4_ (Magnesium sulfate)	0.1
Trace elements *	2 mL/L

* Trace element solution composition (g/L): Na_2_EDTA: 50.0, ZnSO_4_·7H_2_O: 2.2, CaCl_2_: 5.5, MnCl_2_·4H_2_O: 5.06, FeSO_4_: 5.0, CuSO_4_·5H_2_O: 1.57, CoCl_2_·6H_2_O: 1.60.

**Table 3 biology-14-01734-t003:** Analysis of Orthogonal Test Results.

Experiment	A(SDU2)	B(SDU10)	C(SDU20)	Ammonium Removal Efficiency (%)
1	1	1	1	77.31 ± 2.15
2	1	2	2	79.51 ± 1.89
3	1	3	3	92.19 ± 2.03
4	2	1	2	77.90 ± 1.56
5	2	2	3	96.02 ± 1.21
6	2	3	1	80.95 ± 1.78
7	3	1	3	85.08 ± 1.92
8	3	2	1	77.55 ± 2.34
9	3	3	2	84.14 ± 1.67
K_1_	249.01	240.29	235.81	—
K_2_	254.87	253.08	241.55	—
K_3_	246.77	257.28	273.29	—
k_1_	83.00	80.10	78.60	—
k_2_	84.96	84.36	80.52	—
k_3_	82.26	85.76	91.10	—
R	2.70	5.66	12.49	—

Note: Data are presented as mean ± SD (*n* = 3). K = sum of efficiency for each level; k = average efficiency (K/3); R = range (max k − min k).

**Table 4 biology-14-01734-t004:** Sequencing statistics of 16S rRNA gene hypervariable regions.

Sample	Sequence Count	Base Count (M)	Mean Length (bp)	Min Length (bp)	Max Length (bp)
A1	67,763	28.08	414.42	200	456
A2	67,657	28.03	414.27	206	504
B1	68,840	28.61	415.56	234	503
B2	60,774	25.45	418.77	218	449
C1	63,370	26.44	417.16	317	444
C2	65,704	27.81	423.33	206	442
D1	74,078	30.57	412.64	280	502
D2	73,079	30.38	415.72	295	504
Total	541,265	225.37	416.48	200	504

**Table 5 biology-14-01734-t005:** Alpha diversity indices of microbial communities (mean ± SD, *n* = 3).

Sample	Shannon Index	Simpson Index	ACE Index	Chao1 Index	Coverage	Smith-Wilson Index
A1	3.92 ± 0.15	0.126 ± 0.01	1231.92 ± 58	1219.14 ± 62	0.995	0.434 ± 0.02
A2	4.05 ± 0.18	0.108 ± 0.01	1189.73 ± 49	1194.48 ± 57	0.996	0.434 ± 0.02
B1	4.88 ± 0.21	0.029 ± 0.003	1304.74 ± 63	1300.43 ± 65	0.994	0.458 ± 0.03
B2	4.53 ± 0.19	0.037 ± 0.004	1184.10 ± 52	1197.27 ± 59	0.994	0.462 ± 0.03
C1	3.30 ± 0.14	0.082 ± 0.007	924.04 ± 41	721.48 ± 38	0.995	0.495 ± 0.02
C2	3.43 ± 0.16	0.061 ± 0.005	567.13 ± 34	488.50 ± 31	0.997	0.521 ± 0.03
D1	3.33 ± 0.12	0.128 ± 0.01	785.43 ± 35	808.68 ± 41	0.997	0.504 ± 0.02
D2	3.06 ± 0.09	0.137 ± 0.01	577.55 ± 29	563.89 ± 33	0.998	0.510 ± 0.02

## Data Availability

The data presented in this study are available upon request from the corresponding author. The data are not publicly available because of companies’ policies.

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
