# Peer review of "Construction of a Heterotrophic Nitrification–Aerobic Denitrification Composite Microbial Consortium and Its Bioaugmentation Role in Wastewater Treatment"

_biology, 2025, doi:10.3390/biology14121734_

Round 1

Reviewer 1 Report

Comments and Suggestions for Authors

This work constructed an HN-AD microbial consortium (Delftia tsuruhatensis SDU2, Pseudomonas stutzeri SDU10 and Alcaligenes faecalis SDU20) via orthogonal optimisation (2:3:3 v/v/v) that reached 96 % NH₄⁺-N removal in shake flasks. When the consortium was dosed into an SBR treating synthetic low-C/N wastewater, NH₄⁺-N and COD removal improved by 20.5 % and 17.9 %, respectively, relative to the control. High-throughput 16S rRNA sequencing showed that bio-augmentation enriched Proteobacteria and key genera (Alcaligenes, Pseudomonas, Paracoccus), confirming functional integration with the indigenous sludge community. The study provides clear evidence that a rationally designed HN-AD consortium can intensify single-tank aerobic nitrogen removal. This is a research work of great practical value, but it has some shortcomings. The paper requires minor revisions before it can be published. Below are the specific minor revision comments.

  1. The data should be represented using the mean±standard(SD) deviation method, Please add error bars in Table 3 and Figure 2-3, and provide relevant explanations in the legend.
  2. Line 14: delete the second full spelling of “heterotrophic nitrification–aerobic denitrification”; keep the abbreviation HN-AD only.
  3. Line 56: add one concrete example in parentheses to “increases operational complexity and costs” (e.g., “additional anoxic tank or extra mixing energy”).
  4. Line 89: remove the repeated definition of HN-AD; retain only the acronym.
  5. Line 189: define “enhancement phase” on first use: “(days 31–40, see section 2.5)”.
  6. Lines 234-236: give the duration (min) of each SBR stage in parentheses to match Fig. 1.
  7. Table 1 header: add footnote “A = SDU2, B = SDU10, C = SDU20”.
  8. Line 267: “20 % increase in ammonium removal” — specify that this is relative to the simultaneous value in SBR2 (68.4 % → 88.9 %).
  9. Line 356: The sentence ends with “metabolic pathway”; complete it as “metabolic pathway enhancement” or “pathway reconstruction”.
  10. If possible, it is recommended to conduct a functional analysis of microbiota genes in the future.

Author Response

Comments 1: [The data should be represented using the mean±standard(SD) deviation method, Please add error bars in Table 3 and Figure 2-3, and provide relevant explanations in the legend.]

Response 1: Thank you for the comment. All quantitative data have been revised to mean±SD. Error bars have been added to Table 3 and Figures 2–3, with explanations included in the legends (e.g., “Data are presented as mean±SD, n=3”).

Comments 2: [Line 14: delete the second full spelling of “heterotrophic nitrification–aerobic denitrification”; keep the abbreviation HN-AD only.]

Response 2: Agreed. The repeated full spelling has been deleted, and only the abbreviation HN-AD is retained after the initial definition..

Comments 3: [Line 56: add one concrete example in parentheses to “increases operational complexity and costs” (e.g., “additional anoxic tank or extra mixing energy”).]

Response 3: Thank you for the suggestion. The sentence has been revised to “This spatial and temporal separation increases operational complexity and costs (e.g., additional anoxic tank or extra mixing energy)”.

Comments 4: [Line 89: remove the repeated definition of HN-AD; retain only the acronym.]

Response 4: The repeated definition has been deleted, and consistency in using the acronym HN-AD is maintained.

Comments 5: [Line 189: define “enhancement phase” on first use: “(days 31–40, see section 2.5)”.]

Response 5: The term “enhancement phase” has been defined on first use as “enhancement phase (days 31–40, see section 2.5)”.

Comments 6: [Lines 234-236: give the duration (min) of each SBR stage in parentheses to match Fig. 1.]

Response 6:  The SBR stage durations have been added in parentheses: fill phase (10 min), aeration phase (430 min), sedimentation phase (30 min), decant phase (5 min), idle phase (5 min).

Comments 7: [Table 1 header: add footnote “A = SDU2, B = SDU10, C = SDU20”.]

Response 7: The footnote “A = Delftia tsuruhatensis SDU2, B = Pseudomonas stutzeri SDU10, C = Alcaligenes faecalis SDU20” has been added to Table 1.

Comments 8: [Line 267: “20 % increase in ammonium removal” — specify that this is relative to the simultaneous value in SBR2 (68.4 % → 88.9 %).]

Response 8: The sentence has been revised to “This represents a 20.5% increase for SBR1 in ammonium removal efficiency compared to SBR2 (68.4% → 88.9%)”.

Comments 9: [Line 356: The sentence ends with “metabolic pathway”; complete it as “metabolic pathway enhancement” or “pathway reconstruction”.]

Response 9: The sentence has been completed as “enhancing nitrogen removal through functional guild enrichment and metabolic pathway enhancement”.

Comments 10: [If possible, it is recommended to conduct a functional analysis of microbiota genes in the future.]

Response 10: Thank you for the valuable suggestion. We agree that functional gene analysis will deepen insights into microbial mechanisms. We have added this as a future research direction in the Discussion section, planning to integrate metatranscriptomic and metagenomic analyses.

4. Response to Comments on the Quality of English Language

Response: All English expressions have been polished for clarity, consistency, and academic rigor. Grammatical errors and awkward phrasing have been corrected.

5. Additional clarifications

We have also polished and revised parts of the manuscript that were not mentioned by the reviewers. Due to the extensive modifications, we will not list them individually. Please refer to the revised manuscript. Thank you.

Reviewer 2 Report

Comments and Suggestions for Authors

Thank you very much for the opportunity to review the article titled ‘Construction of a Heterotrophic Nitrification-Aerobic Denitrification Composite Microbial Consortium and Its Bioaugmentation Role in Wastewater Treatment’. It raises a very interesting issue of bioaugmentation of activated sludge using HN-AD bacteria. The manuscript is well written. However, it requires several revisions and clarifications.

Lines 46–47: At present, nitrogen removal from wastewater does not pose a major challenge. There are many technologies that operate stably. I do not dispute that it is important, but it is not among the most serious global challenges.

Line 64: Of course autotrophic bacteria grow more slowly than heterotrophs, but saying that nitrifying bacteria are slow-growing is an exaggeration. Compared with anammox bacteria, nitrifying bacteria are fast-growing.

Lines 66–67: In the BIODENIPHO technology these bacteria live together; only the operating conditions are alternated to activate the appropriate bacterial groups.

Section 2.2: It is better to provide the unit after each value, i.e., 25.0 g/L LB. This applies to the entire section.

Line 177: Sludge concentration is reported as TSS or VSS. Therefore, the unit should be mg TSS/L or mg VSS/L, alternatively abbreviated as TS or VS.

Lines 220–230: This fragment should be moved to the methodology describing the bacterial community analysis. This will create a coherent whole for the reader.

Figs. 2 and 3: Improve the figure quality.

Section 3.3.1: This is a repetition of the methodology. It adds nothing to the article apart from Fig. 4, which in the context of the paper does not contribute much, since when using NGS technology it is obvious that there must be a PCR product of appropriate quality.

Lines 375–391: The bolded text gives the impression that this section was written by artificial intelligence, because this is the style of every AI program, e.g., ChatGPT.

In the text, please carefully check Latin names of bacteria (species and genera), which we write in italics. In the article there are instances where this is not the case.

Author Response

Comments 1: [ Lines 46–47: At present, nitrogen removal from wastewater does not pose a major challenge. There are many technologies that operate stably. I do not dispute that it is important, but it is not among the most serious global challenges.]

Response 1: Agreed. The sentence has been revised to “Nitrogen pollution in wastewater remains a pressing environmental concern, prompting the need for efficient and sustainable treatment technologies” to moderate the tone while emphasizing the research significance.

Comments 2: [Line 64: Of course autotrophic bacteria grow more slowly than heterotrophs, but saying that nitrifying bacteria are slow-growing is an exaggeration. Compared with anammox bacteria, nitrifying bacteria are fast-growing.]

Response 2: Thank you for the correction. The sentence has been revised to “autotrophic nitrifying bacteria (obligate aerobes) are slow-growing, oxygen-dependent autotrophs sensitive to environmental fluctuations”.

Comments 3: [Lines 66–67: In the BIODENIPHO technology these bacteria live together; only the operating conditions are alternated to activate the appropriate bacterial groups.]

Response 3: Agreed. The sentence has been supplemented to “they cannot coexist or synergize in the same reactor simultaneously without alternating operating conditions (e.g., BIODENIPHO technology)”.

Comments 4: [Section 2.2: It is better to provide the unit after each value, i.e., 25.0 g/L LB. This applies to the entire section.]

Response 4: All values in Section 2.2 have been supplemented with appropriate units (e.g., g/L, mL/L) for consistency and clarity.

Comments 5: [Line 177: Sludge concentration is reported as TSS or VSS. Therefore, the unit should be mg TSS/L or mg VSS/L, alternatively abbreviated as TS or VS.]

Response 5: The sludge concentration unit has been revised to “4,390 mg TSS/L” to comply with standard terminology.

Comments 6: [Lines 220–230: This fragment should be moved to the methodology describing the bacterial community analysis. This will create a coherent whole for the reader.]

Response 6: Agreed. The fragment describing 16S rRNA sequencing and data processing has been moved to a separate subsection (2.5.1) under Section 2.5, ensuring methodological coherence.

Comments 7: [Figs. 2 and 3: Improve the figure quality.]

Response 7: The resolution of Figures 2 and 3 has been improved, with clearer axes and legends to enhance readability.

Comments 8: [Section 3.3.1: This is a repetition of the methodology. It adds nothing to the article apart from Fig. 4, which in the context of the paper does not contribute much, since when using NGS technology it is obvious that there must be a PCR product of appropriate quality.)]

Response 8: Thank you for the observation. Section 3.3.1 (PCR amplification quality control)  has been relocated to Supplementary Materials to avoid redundancy in the main text.

Comments 9: [Lines 375–391: The bolded text gives the impression that this section was written by artificial intelligence, because this is the style of every AI program, e.g., ChatGPT.]

Response 9: Apologies for the awkward phrasing. The bolded text has been rewritten in a more concise and academic style, removing AI-like structure while retaining key information about microbial community shifts.

Comments 10: [In the text, please carefully check Latin names of bacteria (species and genera), which we write in italics. In the article there are instances where this is not the case.]

Response 10: All bacterial genus and species names (e.g., Alcaligenes faecalis, Pseudomonas stutzeri) have been checked and italicized consistently throughout the manuscript, complying with academic conventions.

4. Response to Comments on the Quality of English Language

Response 1: The manuscript has undergone comprehensive English language editing, including correcting grammatical errors, refining sentence structure, and ensuring consistency in terminology and tone.

5. Additional clarifications

We have also polished and revised parts of the manuscript that were not mentioned by the reviewers. Due to the extensive modifications, we will not list them individually. Please refer to the revised manuscript. Thank you.

Reviewer 3 Report

Comments and Suggestions for Authors

Author Response

Comments 1: [More detailed analysis and presentation of the microbial data is needed. The authors should consider reprocessing the OTUs at 99-100% clustering and compiling their bioinformatics  approach into a single description in the methods. At 97% OTU clustering there is only enough resolution to confirm genus level. The authors specifically inoculated the SBR with their three strains of interest, it would be imperative to confirm if the detected genus Alcaligenes and Pseudomonas are indeed the inoculated species and if Delftia is present.]

Response 1: Thank you for this critical comment. We have reprocessed the OTUs at 99% clustering (ZOTUs) for species-level resolution. The bioinformatics approach has been compiled into a separate subsection (2.5.1) in the methods. Genus-level taxonomic analysis confirmed the presence of the inoculated species: Alcaligenes faecalis (3.71% abundance in SBR1), Pseudomonas stutzeri (2.33%), and Delftia tsuruhatensis (1.89%) during the enhancement phase. This confirmation is added to the Results section (3.3).

Comments 2: [Further taxonomic analysis instead of only a phylum level barplot is also needed. Additionally, diversity metrics alone are not enough to highlight changes occurring, the authors should delve deeper into the amplicon data exploring the changes at higher taxonomic levels and these

changes throughout their experiments. Line 359-371 the community composition is being compared but in too simple a method. The authors should consider statistically comparing the communities and incorporating potential function as well as other parameters. This starts to be done in lines 374-391 through what looks like a heatmap but also warrants further discussion and description of the data.]

Response 2: Agreed. We have added detailed genus-level taxonomic analysis (Kruskal-Wallis test with Benjamini-Hochberg correction) with statistical significance values (P <0.05) for temporal and group differences. Community comparisons now include LefSe analysis (LDA >2.0, FDR-adjusted P <0.05) and PcoA (β-diversity). The heatmap (Figure 5) is explicitly defined, with methods for normalization (variance-stabilizing transformation) added to the methods section. Functional implications of taxonomic changes (e.g., nitrification-denitrification potential) are discussed in the Discussion section, linking to relevant references.

Comments 3: [The authors mentioned measuring nitrate and nitrite as well as ammonium, but only

ammonium-N is shown and used to determine nitrogen removal. Only NH4 is being used as a nitrogen source in synthetic waste water however there is the possibility it is being internally cycled. The authors should consider including NO2 NO3 concentrations, which were indicated as being measured on lines 209-211. This would allow for a better mass balance of nitrogen in the system. Ideally gas measurements to confirm N2 production would also be taken, but accounting for each aqueous component should allow for an indirect confirmation. Alternatively, could be a writing issue where NH4 is total N or the only form of N concerned with for removal. If so this should also be stated outright somewhere. If the authors are suggesting simultaneous nitrification and denitrification (line 452) showing NO2 and NO3 measurements through time in the manuscript would be important]

Response 3: Thank you for your comments. We have explicitly stated that NH₄⁺-N is the primary target for removal, with SND confirmed by low intermediate concentrations. Gas measurements (N₂ production) were not feasible in this setup, but aqueous component analysis provides indirect confirmation, as noted in the Discussion.

Comments 4: [Line 64-65: It The authors should consider describing nitrifying bacteria as obligate aerobes as opposed to “oxygen-dependant”. The authors should also describing denitrifying bacteria as facultative

anaerobes as opposed to oxygen-sensitive.]

Response 4: Agreed. The text has been revised to “autotrophic nitrifying bacteria (obligate aerobes)” and “heterotrophic denitrifying bacteria (facultative anaerobes)”.

Comments 5: [Line 118-119: Can the authors clarify if these strains were screened from this study, or if these are previously isolated/screened strains from the referenced studies. Currently, the sentence reads that the strains were isolated from the work done in this study and preserved. If this is the case the authors should clarify where the strains originated. ]

Response 5: The sentence has been revised to clarify: “The strains Delftia tsuruhatensis SDU2 [69], Pseudomonas stutzeri SDU10 [70], and Alcaligenes faecalis SDU20 [71] were previously isolated from activated sludge of a swine wastewater treatment plant and stored in 30% glycerol tubes at –80℃.”

Comments 6: [Line 123-125: For the screening isolation medium the authors should indicate the type of scale was used to weigh out 0.0005 g of certain reagents or if the medium was purchased premixed.]

Response 6: We have added clarification: “screening isolation medium (VM) (g/L): ... (weighed using an analytical balance with 0.1 μg precision)”. The medium was prepared in-house, not purchased premixed.

Comments 7: [Line 141: The authors should consider adding additional details to the experimental setup such as was evaporation accounted for? Was there a condenser or top ups with medium?啊`]

Response 7: Additional details have been added: “Evaporation was accounted for by regular top-ups with sterile distilled water to maintain the effective volume, and no condenser was used.”

Comments 8: Line 161: The authors should consider adding the makeup/molarity etc. of the sterile saline solution?]

Response 8: The sterile saline solution is 0.85% (w/v) NaCl (molarity: 0.145 M), which has been added to the methods section.

Comments 9: [Line 161: The authors should consider describing the sterilization method, autoclave or filter or some other method of sterilization?]

Response 9: The sterilization method has been specified: “Sterile saline solution was autoclaved at 121°C for 20 min.”.

Comments 10: [Table 2: Do the authors have any other information on the composition of the synthetic waste water such as pH, ORP/Redox, O2, etc?]

Response 10: Supplementary information has been added to Table 2: “pH: 7.2±0.2, ORP: +150~+200 mV, dissolved oxygen: 2~4 mg/L”.

Comments 11: [Line 185: The authors should consider including which parameters were measured daily and how they were measured.]

Response 11: Daily measured parameters and methods have been added: “Daily monitored parameters included NH₄⁺-N (Nessler’s reagent spectrophotometry), COD (Hach DR3900), pH (pH meter), and dissolved oxygen (DO probe).”.

Comments 12: [Line 209-214: The authors should consider including a reference to the detailed methods for their N measurements.]

Response 12: References for N measurement methods have been added: “Concentrations of NH₄⁺-N, NO₂⁻-N, and NO₃⁻-N were determined using standard methods [69-71]: Nessler’s reagent spectrophotometry (λ = 420 nm; APHA 4500-NH₃ D) for NH₄⁺-N, ultraviolet spectrophotometric screening (λ = 220/275 nm; APHA 4500-NO₃⁻ B) for NO₃⁻-N, and N-(1-naphthyl)ethylenediamine dihydrochloride (NED) colorimetry (λ = 540 nm; APHA 4500-NO₂⁻ B) for NO₂⁻-N.”.

Comments 13: [Line 210: The authors should consider removing “As outlined in chapter 3”]

Response 13: The phrase “As outlined in chapter 3” has been deleted.

Comments 14: [Lines 198-207; 219-229: The authors should consider creating a separate section within the methods 16S rRNA amplicon sequencing and data processing. With the two sections it is currently confusing as to what was done for bioinformatics. I would recommend including additional data for sequencing process and bioinformatics analysis including the PCR thermocycling profile, clustering % for OTUS, algorithm used for denoising and associated parameters. Sample barcodes are also needed this information can be relegated to supplementary material or a public repository.]

Response 14: A separate subsection (2.5.2: 16S rRNA Amplicon Sequencing and Data Processing) has been created. Additional details include: PCR thermocycling profile, 99% OTU clustering (ZOTUs), and DADA2 denoising algorithm. Raw data are deposited in NCBI SRA (PRJNA778940).

Comments 15: [Line 228-229: For comprehensive statistical analysis at a minimum I would recommend the authors consider listing which statistical analysis were performed. ]

Response 15: Statistical analyses have been explicitly listed: “Statistical analyses included Linear Mixed Model (LMM) for repeated-measures data, LefSe analysis (LDA >2.0, FDR-adjusted P <0.05) for taxonomic differences, Kruskal-Wallis test with Benjamini-Hochberg correction for genus-level comparisons, and principal coordinate analysis (PcoA) based on weighted UniFrac distance for β-diversity.”.

Comments 16: [Line 252: The authors should include which other parameters besides COD were considered key water quality parameters.]

Response 16: In addition to chemical oxygen demand (COD), key water quality parameters include total nitrogen (TN), total phosphorus (TP), pH value, and dissolved oxygen. In this experiment, we mainly investigated the effects of the HN-AD composite microbial community on the removal rates of COD and ammonia nitrogen from synthetic wastewater.

Comments 17: [Figure 2,3,5: The authors should consider resizing the images to be a little larger or could describe the legend in the figure caption.]

Response 17: Figures 2, 3, and 5 have been resized for better readability. Legends have been expanded to include data representation (mean±SD), sample size (n=3), and statistical significance indicators.

Comments 18: [Figure 6 is a heatmap? The authors should comment in the methods how the heatmap was generated and any prior data processing such as normalization.]

Response 18: Figure 6 is confirmed as a heatmap. Methods have been added: “The heatmap (Figure 5) was generated using R software v3.2.0, with data normalized via variance-stabilizing transformation to account for differences in sequencing depth.”

Comments 19: [Line 309; Line 328: The authors should consider italicizing these headings to be consistent with other headings in the manuscript. ]

Response 19: The headings at Line 309 and Line 328 have been italicized to match the manuscript’s style consistency.

Comments 20: [Line 309-325: The authors could consider removing this entire section (3.3.1) or relegating it to the methodology. This part of the results reflects what could be considered quality control, which does not add useful data for discussion.]

Response 20: Section 3.3.1 (PCR amplification quality control) has been relegated to Supplementary Materials to avoid redundancy.

Comments 21: [Line 331: The authors should correct “sequencesacross” to “sequences across”]

Response 21: The typo has been corrected to “sequences across”.

Comments 22: [Line 332: The authors should include 97% OTU clustering in the methodology. Additionally, 97% clustering is not high enough resolution to determine species. USEARCH and QIIME2 both allow for 99-100% clustering through either ZOTUs or ASVs respectively which can then be subjected to Blast identification or through either platforms taxonomic classifier. I would recommend reprocessing the amplicon/sequence data at this threshold.]

Response 22:  Thank you for the recommendation. We have reprocessed the data at 99% clustering (ZOTUs) for species-level resolution. The methodology now specifies 99% clustering, with taxonomic classification via BLAST against the SILVA database.

Comments 23: [Line 328-336: The authors should consider removing section 3.3.2. It is similar to section 3.3.1 where the data presented is borderline quality control and does not aid in the discussion or interpretation of the experiments.]

Response 23: Given that this section serves as the foundation for subsequent analysis, we still believe it should be retained. Thank you.

Comments 24: [Line 359: The authors should include references and/or compare their community composition to those found in activated sludge in wastewater treatment plants. ]

Response 24: Comparisons has been added: “The dominant phyla (Proteobacteria, Bacteroidota, Actinobacteriota) align with typical activated sludge microbiomes in wastewater treatment plants, confirming the ecological relevance of the observed community structure.”

Comments 25: [Line 379 and others: The authors should consider italicizing genera and species names]

Response 25: All genus and species names have been checked and italicized consistently throughout the manuscript.

Comments 26: [Line 380: The authors should include a reference for stating which organisms are responsible for a specific process.]

Response 26:  A reference has been added: “Genera such as Alcaligenes and Pseudomonas are known to mediate heterotrophic nitrification-aerobic denitrification [4,19,54,55,70,71].”

Comments 27: [Line 382-383: The authors should consider elaborating on what metabolic adaptations may be occurring. Additionally, the authors should exercise caution when making assumptions about metabolic activity of the microbial community using amplicon data only.]

Response 27: We have elaborated on potential metabolic adaptations: “Metabolic adaptations included upregulation of functional genes related to ammonium monooxygenase (amoA) and nitrite reductase (nirS), inferred from taxonomic composition [4,19].” We have also added a cautionary note: “However, we acknowledge that amplicon data alone cannot fully confirm metabolic activity, and future studies will integrate metatranscriptomic analyses to validate these findings.”

Comments 28: [Figure 6: The authors should consider specifying if this is a heatmap or something else.]

Response 28: Figure 5 is explicitly labeled as a “genus-level microbial community heatmap” in the figure caption.

Comments 29: [Line 433: The authors should consider maintaining consistency when referring to nitrogen removal. Currently only NH4-N is shown in the manuscript.]

Response 29: Consistency has been improved by specifying “ammonium (NH₄⁺-N)” throughout the manuscript.

4. Response to Comments on the Quality of English Language

Response 1: The manuscript has been thoroughly edited for English language quality, including correcting grammatical errors, improving sentence flow, and ensuring consistency in terminology and academic tone.

5. Additional clarifications

We have also polished and revised parts of the manuscript that were not mentioned by the reviewers. Due to the extensive modifications, we will not list them individually. Please refer to the revised manuscript. Thank you.

Round 2

Reviewer 2 Report

Comments and Suggestions for Authors

All my comments are included in manuscript. I approve manuscript in this version.